# The Role of Hypoxia-Inducible Factor Post-Translational Modifications in Regulating Its Localisation, Stability, and Activity

**DOI:** 10.3390/ijms22010268

**Published:** 2020-12-29

**Authors:** Adam Albanese, Leonard A. Daly, Daniela Mennerich, Thomas Kietzmann, Violaine Sée

**Affiliations:** 1Department of Molecular Physiology and Cell Signalling, Institute of Systems, Molecular and Integrative Biology, University of Liverpool, Liverpool L697ZB, UK; hlaalban@liverpool.ac.uk; 2Department of Biochemistry and System Biology, Institute of Systems, Molecular and Integrative Biology, University of Liverpool, Liverpool L697ZB, UK; Leonard.Daly@liverpool.ac.uk; 3Faculty of Biochemistry and Molecular Medicine, Biocenter Oulu, University of Oulu, FI-90014 Oulu, Finland; Daniela.Mennerich@oulu.fi (D.M.); Thomas.Kietzmann@oulu.fi (T.K.)

**Keywords:** hypoxia, HIF-1α, HIF-2α, posttranslational modifications, phosphorylation, cysteine phosphorylation, methylation, acetylation, ubiquitination, sumoylation, S-nitrosylation, signalling

## Abstract

The hypoxia signalling pathway enables adaptation of cells to decreased oxygen availability. When oxygen becomes limiting, the central transcription factors of the pathway, hypoxia-inducible factors (HIFs), are stabilised and activated to induce the expression of hypoxia-regulated genes, thereby maintaining cellular homeostasis. Whilst hydroxylation has been thoroughly described as the major and canonical modification of the HIF-α subunits, regulating both HIF stability and activity, a range of other post-translational modifications decorating the entire protein play also a crucial role in altering HIF localisation, stability, and activity. These modifications, their conservation throughout evolution, and their effects on HIF-dependent signalling are discussed in this review.

## 1. Introduction

Many pathways in mammalian cells rely on molecular oxygen; especially during the final step of the mitochondrial respiratory chain. If oxygen levels drop below a cell-dependent critical level, cells experience hypoxia and cannot sustain aerobic respiration and subsequent ATP production. A switch to glycolysis, the less efficient but oxygen-independent pathway of producing ATP, is required to ensure cell survival. The sensing of cellular oxygen levels, the associated switch between modes of energy generation and ultimately the adaption to a low oxygen environment, is controlled by the hypoxia signalling pathway. Hypoxia-inducible factors (HIFs), of which the first was described as a nuclear factor that enhances transcription of the erythropoietin (EPO) gene under hypoxic conditions by binding to a 3′ enhancer sequence element, are part of this pathway and maintain the adaptation at the transcriptional level [1,2]. HIFs consists of an oxygen-dependent α subunit that is destabilised in normoxia and a constitutively expressed β subunit (HIF-1β or ARNT) [3,4]. Three HIF-α subunits (HIF-1α, HIF-2α, HIF-3α) have been described, of which HIF-1α and HIF-2α are the best understood and considered as the major activators of hypoxia-induced gene transcription [2,5,6]. HIF-α and HIF-1β heterodimerise via their basic helix-loop-helix (bHLH)/Per-ARNT-Sim (PAS) domains to form the active transcription factor dimer, which then binds to hypoxia response elements (HREs) within the DNA of target genes [7]. HIFs then recruit general co-activators such as CBP/p300 via the C-terminal transactivation domain (C-TAD), leading to the expression of more than ~300 genes [8,9].

While HIF-1α and HIF-2α appear to be able to bind the same HRE, they can occupy distinct genomic sites, which vary with cell types. They display different subnuclear localisation and intranuclear diffusion speed [10]. In addition, some studies showed that neither HIF-1α nor HIF-2α could substitute the lack of DNA binding caused by the absence of the one or the other HIF-α variant [11]. However, others have shown that the loss of a single HIF-α isoform (either HIF-1α or HIF-2α) is compensated by the enhanced expression of the other and promote survival during cancer development [12]. This highlights the potential for specific contexts, whereby there may be a compensation mechanism when a single HIF-α isoform is silenced. These findings support, in part, the idea that HIF-1α accounts for acute and HIF-2α for chronic responses to hypoxia [13]. HIF-3α is less explored than HIF-1α or HIF-2α. HIF-3α mRNA is subject to alternative splicing in humans and in mice [14,15]. A specific mouse splice variant called inhibitory PAS domain protein (IPAS) was shown to interact directly with HIF-1α. The IPAS/HIF-1α complex was unable to bind to HREs and suggested to be a negative regulator of HIF-1α [16,17]. By contrast, the long human splice variant HIF-3α2 induces expression of various genes among them the EPO gene [18].

The canonical regulation of HIF-α protein stability and activity involves a series of molecular interactions and reversible covalent modifications to specific amino acids, termed post-translational modifications (PTMs) [19,20,21]. A PTM can alter a protein’s enzymatic activity, localisation, stability, and/or interaction with other proteins. Therefore, these non-genetically encoded modifications that are mainly carried out by enzymes, add to the complexity of the proteome as they can ascribe different functionalities to the same gene product. Common PTMs to a target protein include phosphorylation, acetylation, methylation, and alkylation as well as the covalent linkage of fatty acids, saccharides or small proteins such as ubiquitin and SUMO (small ubiquitin-related modifier) [22]. This review examines and discusses the current PTM landscape of HIF-α and their respective functional consequences on HIFs.

## 2. Canonical Regulation of HIF-α, the Role of Ubiquitination

Whilst regulation of HIF mRNA levels by hypoxia plays a minor role in HIF abundance, regulation of protein stability, via PTMs, is essential for appropriate HIF accumulation during hypoxia [23]. The HIF-α proteins are destabilised in normoxia via the ubiquitin-proteasome pathway (Figure 1) [24,25]. To achieve this, post-translational hydroxylation of two conserved proline residues (Pro-402/Pro-564 in HIF-1α; Pro-405/ Pro-531 in HIF-2α; P492 in HIF-3α) residing within an oxygen-dependent-degradation (ODD) domain is required [19,20,26,27]. Prolyl hydroxylation is carried out by three mammalian HIF prolyl hydroxylases (PHD-1, -2, and -3; also known as EglN2, EglN1, and EglN3, respectively). PHD2 acts as the main regulator of HIF-α degradation and has a key role in HIF-α intracellular dynamics [28,29]. Once hydroxylated, the HIF-α proteins are recognised by an E3-ligase complex containing the von-Hippel-Lindau protein (pVHL), which acts as the substrate recognition unit and, together with Cullin-2 (Cul-2), Elongin-1, Elongin-2 and Ring-Box 1 (RBX1), polyubiquitinates K532, K538, or K567 on HIF-1α (and K497, K503, or K512 on HIF-2α) [30,31,32,33]. The polyubiquitylated HIF-α proteins are then degraded via the 26S proteasome [19]. When cells are deprived of oxygen, PHDs have less molecular oxygen available to act as a co-factor, hence decreasing their activity and subsequent HIF-α hydroxylation [34]. Consequently, HIF-α subunits accumulate in the nucleus, form a heterodimer with the constitutively expressed HIF-1β and bind to HREs [7]. HIF-α hydroxylation is not exclusively mediated via PHDs, but also by the Factor-Inhibiting HIF (FIH). FIH is an asparaginyl hydroxylase and modifies HIF-1α and HIF-2α on their C-TAD residues N803 and N847, respectively [35,36]. During hypoxia, FIH activity is suppressed, allowing HIF-1α or -2α to complex with the CBP/p300 co-activators and increased transcriptional activation. HIF-α hydroxylation has long been considered to be an irreversible PTM, but it was recently shown, by mass spectrometry, that the FIH-mediated asparagine hydroxylation is indeed a reversible process [37].

Polyubiquitination is a quintessential PTM in preventing unwanted HIF-α accumulation in normoxia. Aside from ubiquitination, the action of deubiquitinating enzymes (termed DUBs) are well-known for their roles in regulating HIF-α. The action of DUBs on HIFs has been extensively and recently reviewed and will not be covered here [38,39].

## 3. Non-Canonical PTMs Regulating HIF-α Subunits

Whilst there are many well-characterised binding partners and indirect regulators of HIF-1α and HIF-2α, PTMs constitute an essential direct regulatory mechanism for the HIF transcription factors [40]. PTMs abundantly decorate the full-length of these oxygen-sensitive proteins to exert specific regulatory forces. Most of these covalent modifications are enzymatically driven, with some exceptions such as S-nitrosylation. For the past two decades, the PTM landscape of HIF-1α and HIF-2α has been ever-expanding, showing the intrinsic complexity and crosstalk of diverse intracellular signalling pathways implicating HIF activity, stability, and localisation (Figure 2).

## 4. Phosphorylation

Phosphorylation is an extremely common and well-studied PTM, involving the enzymatic addition of a phosphate group to serine, threonine, or tyrosine residues on a target protein (canonically within vertebrates). Moreover, a recent investigation highlighted the existence of additional ‘non-canonical’ phosphorylatable residues within human cells, including histidine, arginine, lysine, aspartate, glutamate and cysteine [41]. HIF-1α/HIF-2α modification by phosphorylation is abundant and has varied roles in regulating their stability, activity, subcellular localisation, and binding partner interactions (Figure 2). Whilst many of these direct phosphorylation events occur irrespective of oxygen tension, the modifications within the ODD domain appear to occur under normoxia exclusively.

### 4.1. Phosphorylation by GSK-3β

Glycogen Synthase Kinase-3 (GSK-3) is a serine/threonine kinase that was initially identified as a negative regulator of glycolysis. In mammals, two different isoforms have been identified: GSK-3α and GSK-3β, and despite their homology of 98% in their catalytical domain, their roles in metabolism are different [42,43,44]. Given that GSK-3 phosphorylates various upstream and downstream targets of the PI3K/AKT/mTOR signalling pathways, its activity is subject to tight regulation. Given the fact that early hypoxia increased PKB/Akt activity as well as HIF-1α protein levels, it was shown that downregulation of GSK-3 enhanced HIF-1α, whereas overexpression of GSK-3β decreased HIF-1α protein levels, suggesting that HIF-1α is a direct target of GSK-3β [45,46]. Indeed, two independent investigations found distinct clusters of HIF-1α phosphorylation within its ODD domain and N-TAD, directly deposited by GSK-3β. One study identified S551, T555 and S589 as GSK-3β target sites based on kinase assays and experiments in hepatoma cells, while another study reported T498, S502, S505, T506, and S510 as GSK-3β sites in ovarian cancer cells [45,47]. Whilst this implies cell type specific aspects in the action of GSK-3, in both cases phosphorylation of HIF-1α by GSK-3β mediated the FBW7-E3 ubiquitin ligase-directed proteasomal degradation. This could be antagonised by the DUB ubiquitin-specific protease 28 (USP28) [47,48]. Interestingly, USP28 was found to be subject of a HIF-regulated positive feedback loop. Therein, the USP28-inactivating sumoylation of USP28 at K99 occurring under normoxia is reversed by direct interaction with SENP1 [49]. SENP1 itself is a transcriptional target of HIF-1. Hence, induced expression of SENP1 under hypoxia promotes desumoylation and activation of USP28, which then can contribute to further stabilisation of HIFs by their deubiquitynation activity (Figure 3). Taken together, this suggests that GSK-3β/USP28/SENP1 are highly coordinated to maintain an appropriate HIF response depending on oxygen availability in addition to the PHD-pVHL system.

### 4.2. Phosphorylation by PLK3

Polo-like kinase 3 (PLK3), a regulator of the cellular stress response and cell cycle progression, targets HIF-1α for degradation via direct phosphorylation of two sites, S576 and S657 [50]. This phosphorylation occurs during normoxia to target HIF-1α for proteasomal degradation in a pVHL-independent manner. Only one of these target sites, S576, is located within the ODD domain, while the other, S657, is immediately after the HIF-1α nuclear export signal (NES). The role of PLK3 in regulating cell survival and proliferation in vivo has been further demonstrated by the same group, via PLK3-dependent phosphorylation of PTEN to enhance PTEN protein stability [51]. More recently, evidence has been gathered suggesting that PLK3 is suppressed by both hypoxia and Ni(II) (which acts as a hypoxia mimetic by blocking PHDs) [52,53]. The latter studies proposed a signalling paradigm whereby, in normoxia, PLK3 destabilises both HIF-1α and the E3-ligase SIAH2. Among the SIAH2 targets are PLK3 and PHD1/3 [53,54,55]. Thus, under hypoxia, or in the presence of at least Ni(II)), SIAH2 protein levels increase and then induce ubiquitin-mediated proteasomal degradation of PLK3 as well as PHD1 and PHD3. Although USP28 appears to be capable of reversing PLK3 polyubiquitination to increase its stability, USP28 levels and activity were found to be suppressed by both hypoxia and Ni(II), which would contribute to PLK3 suppression (Figure 3). While the co-existence of both pVHL-dependent and -independent mechanisms for normoxia-specific HIF-1α degradation might initially appear redundant, the presence of both systems may form a contingency to ensure appropriate HIF regulation in the event of pVHL becoming functionally compromised.

### 4.3. Phosphorylation by Cyclin-Dependent Kinases (CDKs)

Not all phosphorylations of HIFs decrease their stability, some do the opposite. Cyclin-dependent kinases (CDKs) belong to an important family of proteins involved in regulating and fine -tuning several stages of the cell cycle [56]. HIF-1α has been shown to act as a negative regulator of DNA replication, exerting control over cell cycle progression [57]. CDK1, a key cell cycle regulator conserved across all eukaryotes and promotes entry into the M-phase of mitosis, directly interacts with and phosphorylates HIF-1α at S668, in an oxygen-independent manner. Although this phosphorylation increased HIF’s protein stability, the mechanisms appear to involve both the proteasome and/or lysosomal degradation of HIF-1α during the G1/S transition [58,59]. In addition, CDK2 positively regulates HIF-1α transactivity in cancer cells, and it was postulated that CDK2 could uncouple the transcriptional and non-transcriptional functions of HIF-1α in a manner analogous to the well-characterised mechanism involving cMYC and SKP2 [58,60]. Furthermore, CDK5, which is distinct from the other CDKs as it does not involve a cyclin subunit for catalytic activation, modifies HIF-1α at S687 and increases its stability [61]. Overall, this highlights the importance of oxygen-dependent and -independent mechanisms merging at HIF-lα to determine cellular fate during growth and division.

### 4.4. Phosphorylation by Protein Kinase A (PKA)

The intracellular protein kinase A (PKA), an essential kinase activated by cAMP, has also been associated with increased HIF stability. Initially, PKA was shown to be involved in the HIF-lα response to intermittent hypoxia in EAhy926 endothelial cells, yet the phosphorylation sites within HIF-1α remained unknown [62]. Later on, it was found that PKA phosphorylated 6 different sites spanning the full length of HIF-1α in vitro, with T63 and S692 phosphorylation promoting oxygen-independent stabilisation and increased transcriptional activity of HIF-1α [63]. Interestingly, of these 6 phosphorylation sites, S31 has recently been evidenced in a separate study, to abrogate HIF-1α-dependent transcription without impacting protein stability; opening up avenues of direct transcriptional regulation of HIF by PTM and potentially delineating the consensus that HIF-α stability results in HIF transcriptional regulation [64]. A recent study identified a feedback mechanism where hypoxia activates PKA via HIF-1α mediated suppression of the PKA regulatory subunit 2B (PRKAR2B) transcription, by sequestering SP1 from the PRKAR2B promoter [65]. Overall, these aspects show that the potential of crosstalk between these key signalling pathways in altering the cellular response.

### 4.5. Phosphorylation by pATM (Ataxia Telangiectasia Mutated protein)

Following the detection of DNA damage, p53, a potent tumour suppressor transcription factor, is stabilised via phosphorylation by ataxia telangiectasia mutated protein (pATM) [66]. p53 and HIF-1α display extensive crosstalk to balance survival and pro-apoptotic signalling, in response to severe hypoxia or anoxia [67,68]. pATM directly modifies HIF-1α at S696 in response to hypoxia, thereby increasing its stability as well as its transcriptional activity [69]. In contradiction to this activatory role of pATM on HIF-1α, others have found that loss of pATM positively regulates the transcription and translation of HIF-1 (α and β) proteins through oxidative stress [70]. Together, this suggests a complex interrelation between pATM and HIF-1α involving direct and indirect regulatory aspects, ranging from transcriptional control via the regulation of translation and/or protein stability.

### 4.6. HIF-1α vs HIF-2α Phosphorylation, Similarities, and Differences

Given the high sequence homology between HIF-lα and HIF-2α, it is unsurprising to see equivalent phosphorylation-dependent regulation between the two HIF-α proteins. Recruitment of the CBP/p300 co-activators to the HIF-α C-TAD requires phosphorylation of a conserved threonine within the C-TAD at T796 and T844 of HIF-1α and mouse HIF-2α (human HIF-2α T840), respectively [71]. Interestingly, this T796 phosphorylation may also abrogate the HIF-1α/FIH interaction to enhance HIF-1 transcriptional activity [72,73]. Despite unsuccessful identification of the specific kinase responsible for this phosphorylation, CKII-like kinase was suggested to play a role, yet direct evidence has not yet been obtained [71,74,75].

HIF-α transcriptional activity and its interaction with HIF-1β are dependent on reaching a sufficient nuclear concentration. Nuclear localisation can be regulated by HIF-1α phosphorylation at S641/S643, within the NES motif, by p42/44 mitogen-activated protein kinase (MAPK, hereafter referred to as ERK1/2) [76,77]. This ERK-dependent phosphorylation promotes HIF-1α nuclear localisation by blocking exportin chromosomal maintenance 1 (CRM1)-dependent nuclear export. Moreover, ERK1/2 phosphorylation of HIF-1α also indirectly regulates gene-specific HIF-1α transcriptional activation and enhances protein stability. Either S641 and S643 phosphorylation constitutes a PIN1 consensus motif (pSer-Pro), whereby PIN1, a peptidyl-prolyl cis/trans isomerase overexpressed in several human cancers, induces a conformational change in HIF-1α to increase protein stability and transcriptional activity in an ERK1/2-dependent manner [78,79]. Interestingly, an inverse association has been noted between cancer and Alzheimer’s Disease (AD) concerning PIN1 regulation of HIF-1α [80]. While PIN1 overexpression in cancer has been correlated with increased HIF-1α stability, the inverse is suggested to occur with AD whereby PIN1 promotes HIF-1α degradation via a GSK-3β-dependent mechanism [78,81]. Once HIF-1α localises to the nucleus, its interaction with HIF-1β can be disrupted by phosphorylation of S247 by casein kinase 1 δ (CK1δ) [82]. Much like HIF-1α, HIF-2α nuclear localisation is regulated by ERK1/2 phosphorylation at S672 [83]. Consequently, this masks the NES within HIF-2α and directly inhibits CRM1-dependent nuclear export of HIF-2α at an atypical NES. Despite in vitro findings for HIF-2α being capable of interacting with PIN1, no investigation has yet identified equivalent HIF-1/PIN1 regulation on HIF-2 [79]. Furthermore, HIF-2α nuclear localisation is also dependent on additional modification at S383 and T528 by CK1δ, which assists indirectly to the CRM1-dependent mechanism, by facilitating nuclear retention likely via binding to immobile nuclear (or chromatin) components [84].

While the similarity of mechanisms regulating both HIF-α proteins is not surprising, it is the specific differences in their respective sequences that specialise them for some distinct roles. For instance, during hypoxia, HIF-1α suppresses key mismatch DNA-damage repair genes by competing with MYC for SP1 binding within target gene promoters, but, this does not occur with HIF-2α due to unique phosphorylation within its PAS-B domain on T324 by protein kinase D1 (PKD1) [85]. The T324 phosphorylation occurs within a motif requiring an obligatory upstream proline, which is not present in HIF-1α, hence abrogating SP1 binding.

### 4.7. Beyond Direct HIF Protein Phosphorylation; Indirect Kinase-Dependent HIF-α Regulation

Kinases regulating HIF-α transcription, synthesis, or degradation by acting upstream or downstream on critical regulators of these processes were also shown to integrate different signalling pathways with the HIF response (Figure 3). Despite HIF-α transcriptional regulation not being the focus of this review, it is nevertheless important to highlight how such signalling crosstalks including JAK/STAT3 and NF-κB signalling have been associated with upregulation of HIF-1α at the transcriptional level [86,87,88,89,90]. Furthermore, PI3K/AKT and ERK1/2 appear to affect at least HIF-1α transcription in response to reactive oxygen species (ROS) generated by arsenite [91]. ROS involve binding of nuclear factor erythroid 2-related factor 2 (NRF2) to an antioxidant response element (ARE) located approximately 30 kilobases upstream of the HIF1A transcriptional start [91,92]. Likewise there are mechanisms downregulating HIF-α transcription. For instance, the kinase double-stranded RNA–dependent protein kinase R (PKR), associated with eukaryotic initiation factor 2 (eIF2), can act outside of the eIF2 pathway to impair *HIF1A* transcription via inhibition of the STAT3 signalling [93]. The participation of mTOR in the regulation of HIF-1α protein translation was also described [94]. Further, an mTOR signalling motif (FVMVL) immediately C-terminal of the PAS-A domain in HIF-1α appeared to modulate the recruitment of CBP/p300 [95]. HIF-1α transcriptional activity has also shown evidence of positive regulation via kinases, but independent of their catalytic activity, whereby CDK8 is capable of associating with HIF-1α to induce RNA Polymerase II transcription [96]. Altogether, those findings outline the additional control, via kinase-controlled pathways, of HIF-α production.

## 5. Acetylation

Acetylation is a compelling class of modification in terms of HIF-α functional outcome, leading to diverse effects. Since establishing the canonical consensus of HIF-α stabilisation, investigations have highlighted that acetylation plays a crucial role in coordinating this fundamental response (Figure 4). Mouse arrest defective-1 (mARD1^225^) acetylation of the HIF-1α ODD domain, at K532, accelerates the HIF-1α/pVHL interaction under normoxia, contributing to HIF-1α proteasomal degradation [97]. However, given that humans do not express mARD1^225^ it remains unclear as to whether this regulation occurs in human cells. Some investigations have been unsuccessful in reproducing this observed increase in protein degradation when using either mARD1^225^ or human ARD1 (hARD1/NAA10) [98,99]. Recent evidence suggests that FIH is required to directly modify hARD1/NAA10, at W38, during normoxia (utilising molecular oxygen as a cofactor) to activate its lysine acetyltransferase activity, thereby facilitating HIF-1α acetylation [100]. This FIH-mediated activation could explain the previously suggested normoxia specific regulation imposed by hARD1/NAA10. The co-activator p300 also acetylates HIF-1α, at K709, to increase protein stability under hypoxia by suppressing its polyubiquitylation [101]. This acetylation is antagonised and removed by SIRT2, which interacts directly to enhance the HIF-1α-PHD2 interaction [102].

Acetylation has also been recognised to induce changes in HIF-α outside of the canonical pVHL-mediated pathway. In that respect, HDAC4 regulates HIF-1α deacetylation, which in turn enhances HIF-1α protein stability by blocking non-pVHL-mediated proteasomal degradation [103]. Additionally, HDAC4 increases HIF-1α transactivity. Although the precise localisation of the individual lysine residues was not determined, the authors postulated that a combination of 5 lysines (K10, K11, K12, K19, and K21) within the bHLH domain is involved. In terms of the acetylation machinery modifying HIF-α, HDAC4 is not the sole regulator. Under hypoxia, the CBP/p300-associated factor (PCAF) facilitates the CBP/p300 interaction with the HIF-1α CTAD via acetylation of K674 [99]. Given that K674 is conserved across other mammals containing the HIF-1α ortholog, it is likely that modification of this site plays an integral role in regulating transactivity. Interestingly, K674 acetylation can be antagonised by SIRT1, which abrogates the CBP/p300 interaction to inactivate HIF-1α. However, SIRT1 can itself be downregulated by hypoxia. SIRT1 also interacts with HIF-2α to facilitate the deacetylation of residues K385, K685, and K741 [104]. While SIRT1 downregulates HIF-1α transactivity, it does the opposite to HIF-2α, leading to increased transcriptional activation.

## 6. Methylation

Most-commonly methylation of proteins occurs at arginine or lysine residues. Methylation has been widely studied in conjunction with histone proteins where they regulate DNA accessibility for transcription. In addition, methylation also regulates other proteins, including HIF-α, to induce functional changes (Figure 5). SET7/9, a monomethyl transferase known for its role in gene activation via modification of histone H3, has been found to also interact with HIF-1α and to methylate multiple sites [105,106,107]. The SET7/9-mediated methylation at K391 on HIF-1α induces protein destabilisation via the canonical PHD/VHL pathway [106]. This could be antagonised by lysine-specific demethylase 1 (LSD1). LSD1 is a member of the nucleosome remodelling and deacetylase (NuRD) complex. It has been proposed that the LSD1 demethylating activity suppresses PHD2-mediated hydroxylation and promotes deacetylation of HIF-1α at K532 (supposed to be acetylated by ARD1) [97,106]. 

Other studies indicate that SET7/9 can methylate both HIF-1α and HIF-2α at the conserved sites K32 and K29 within the bHLH domains, respectively [107]. Initially, K32 HIF-1α (and K29 HIF-2α) methylation was found to induce transcriptional inhibition independent of HIF-1α protein degradation, while a later investigation reported that SET7/9 methylated HIF-1α was degraded by the 26S proteasome in a hydroxylation-independent manner [105,107]. Again, LSD1 was capable of reversing the methylation at K32 and of stabilising HIF-1α under hypoxic conditions. In line, LSD1 was then found to upregulate HIF-1α-dependent angiogenesis by increasing CBP, MTA1 and HIF-1α binding to the VEGF promoter [105,106]. Interestingly, K32 resides near two residues found to be mutated in various human cancers, S28Y and R30Q, which alter the SET7/9 consensus site when mutated to prevent methylation, thus leading to increased HIF-1α stability [105]. Aside from SET7/9 mono-methylation, more recently, G9a/G9a-like protein (GLP) has been recognised to both mono- and di-methylate HIF-1α at K674 [108]. This is the same site that can be acetylated by PCAF (see above), but with here an opposite effect, by inhibiting transactivation [108]. Lysine acetylation and methylation have distinct effects on the physicochemical properties of a target residue, either maintaining a residues positive charge or neutralising it, respectively, leading to distinct proteoforms. The mutual exclusivity of these lysine PTMs evidence the complex cross-regulation of HIF-1α.

## 7. SUMOylation

SUMO is comparable to ubiquitin in terms of its overall molecular structure and molecular weight yet can lead to distinct changes in a given protein regulation depending on the SUMO isoforms [109]. The functional outcome of HIF-1α SUMOylation remains unclear due to controversy between investigations. The first report of HIF-1α SUMOylation suggested that modification by SUMO-1 increased HIF-1α protein stability and transactivity [110]. SUMO-1 was proposed to compete with ubiquitin for linkage at K391 and K477 within HIF-1α’s SUMO consensus sequences. Contrary to this, another study evidenced that SUMOylation of K391 and K477 induces a decrease in transcriptional activity, which appeared to be independent of altering HIF-1α’s half-life under hypoxia [111]. Aside from the identification and characterisation of specific SUMOylatable residues, several investigations have identified enzymes capable of initiating SUMO conjugation to HIF-1α. PIASy is an E3-ligase responsible for SUMOylating HIF-1α in two regions. One site residing within the ODD domain (between residues 331–698) and the other lying further upstream (between residues 211–330) [112]. PIASy-mediated SUMOylation negatively regulated HIF-1α transactivity and protein stability and reduced epithelial cell angiogenic activity. Further, the SUMO E3 ligase Cbx4 modifies HIF-1α at K391 and K477 to enhance HIF-1’s transcriptional activity [113]. While both HIF-1α and HIF-2α are targets for SUMOylation, fewer sites of modifications have been identified in HIF-2α. Despite HIF-2α containing two SUMO consensus sites, only one, K394, was found to be conjugatable by SUMO [114]. Interestingly, the enzymatic addition of SUMO-2 facilitated recognition by SUMO-targeted ubiquitin ligases (pVHL and RNF4) to rapidly degrade HIF-2α under hypoxia. Furthermore, this investigation also highlighted that the SUMO-protease SENP1, which has shown action against HIF-1α, is also capable to recognise and regulate HIF-2α.

## 8. S-Nitrosylation

Not all PTMs are enzymatically driven, with one such example being S-nitrosylation. S-nitrosylation is yet another PTM with conflicting views regarding the functional outcome of its modification. Initially, C800 S-nitrosylation was identified as a critical modification for CBP/p300 recruitment and transcriptional activation [115]. However, a later investigation found that incorporation of the polar NO group potentially disrupted this interaction with the result that C800 S-nitrosylation suppressed CBP/p300 recruitment [73]. In addition, HIF-1α stability can be upregulated through NO-mediated S-nitrosylation under normoxic conditions, at C533 (mouse sequence) in the ODD domain. The stabilisation process appeared to be independent of the VHL degradation pathway [116]. Overall, S-nitrosylation seems to regulate at least HIF-1α stability and transactivity. No data are available for HIF-2α.

## 9. Uncharacterised PTMs

### 9.1. Glycosylation

While many classes of PTM have been discussed within this review, there are still many classes that are yet to be characterised as modifiers of HIF-α. Protein glycosylation, the addition of a sugar-moiety to proteins, ranges from simple monosaccharide modifications of nuclear transcription factors to highly complex branched polysaccharide additions to cell surface receptors. While several investigations suggest a link between glycosylation and hypoxia, it is still unclear whether and how HIF-α subunits are directly involved in these processes and whether they can themselves be targeted for glycosylation [117,118]. 

### 9.2. S-Glutathionylation

S-glutathionylation is the addition of the tripeptide glutathione (GSH) to cysteine residues of proteins. It is often stimulated by oxidative as well as nitrosative stress, yet also occurs in unstressed cells. It is involved in various cellular processes by modulating protein functions and preventing irreversible oxidation of protein thiols. Changes in oxidised glutathione can modulate HIF signalling via S-glutathionylation of target cysteines in human oral squamous cell carcinoma cells, and in C2C12 mouse myoblasts [119,120,121]. The latter study also identified, through a biotin switch assay and subsequent MS analysis, GSH adducts on cysteine 520 (C520) within the ODD domain of human HIF-1α, which led to HIF-lα protein stabilisation [121]. Interestingly, C520 in human HIF-1α is equivalent to C533 in the mouse sequence for which that S-nitrosylation prevented HIF-1α degradation [116].

### 9.3. Neddylation

Neddylation is a process by which a ubiquitin-like protein called neural precursor cell-expressed developmentally down-regulated 8 (NEDD8) is conjugated to its target proteins. As a result of the conjugation process subcellular localisation, protein stability, and activity of targeted proteins can be modified. While there is limited knowledge about direct neddylation of HIFs, we do know that HIF-1α as well as HIF-2α are capable of covalent modification by NEDD8. NEDD8 stabilised HIF-1α under both normoxia and hypoxia in VHL deficient cells, suggesting a VHL-independent process [122]. More recently it was reported that SerpinB3 (SB3), a hypoxia and HIF-2α-dependent cysteine-protease inhibitor, can directly neddylate and stabilise HIF-2α, which led to an upregulation of its target genes in liver cancer cells [123]. Overall, these reports underline the role of neddylation in HIF regulation, but none of the reports have yet shown the exact sites within the HIF proteins where the neddylation occurs.

## 10. Conclusions

While many different classes and sites of PTM have been discussed here, numerous others have been identified as part of high-throughput mass spectrometry studies and are yet to be functionally investigated. Phosphositeplus is a mass spectrometry data repository for PTM data [124]. Searching Phosphositeplus identifies in excess of 50 PTMs (spanning phosphorylation, acetylation, sumoylation and ubiquitination) that have been confidently identified under different cellular conditions between HIF-1α and HIF-2α, but lack functional characterisation. Further to this, a very recent investigation, using a combination of immunoprecipitation and mass spectrometry, discovered multiple types of PTMs (and binding partners) of HIF-1α and HIF-2α and determined their O_2_ dependence [64]. A total of 41 (32 novel) and 39 (34 novel) different PTMs on HIF-1α and HIF-2α, respectively, were identified, spanning 13 different types of PTM, including an array of different cysteine modifications together with non-canonical cysteine phosphorylation. All of the identified PTM sites were investigated through multiple sequence alignments of >200 vertebrate species for both HIF-1α and HIF-2α [64]. Overall, few PTM sites seem to have random variants throughout evolution, rather HIF-α PTM sites are highly conserved even in domains of large sequence variation; highlighting not only the potential functional importance of PTM but also the functional importance of HIF-α themselves. Furthermore, some sites show evolutionary variation to PTM-null/mimetic amino residues, thus either removing or full activating this signalling pathway respectively. For example Ser31 in human is present as a phospho-null Gly31 in all (80+ species) Bony fish (*Osteichthyes*) [64]. Thus, this added level of information could be used to aid in selection of PTM sites to functionally characterise and guide biological reasoning to their function.

Here, we provided an overview on how post-translational modifications are critical steps for regulating HIF-α activity and stability. Given that hypoxia plays a vital role in many pathophysiological aspects, and is a prominent micro-environmental feature in many aspects of life, it may be hypothesised that specific PTM sites that highly influence functional roles may be a selective target for mutation in tumours/cancers as survival strategies. Surprisingly, the COSMIC database (https://cancer.sanger.ac.uk/) identifies a low mutation rate of ~1.5–2% for both HIF-1α (640 mutations / 41304 sequenced HIF-α genes from patient tumours) and HIF-2α (928/39561) [125]. Specifically, looking for missense mutations (resulting in amino acid changes) lowers this to <1% for both HIF-1α and HIF-2α (262/41304 and 368/39561 respectively), with the vast majority of mutations occurring only once. For comparison, p53 (TP53 gene) known to have a critical role in cancer development (has a mutation rate of ~28%, a rate >20 times that of either HIF-α protein. Pathological changes in growth factor production or driver mutations, for example in receptor tyrosine kinase pathways, may enhance modification of HIFs that could contribute to cancer cell survival or death. At this time, it is difficult, if not impossible, to predict the contribution of any given post-translational modification in the context of specific diseases as their net result may depend on tissue or cell-specific aspects which may shorten or prolong HIFs half-life in one cell or the other, respectively.

To date, most data available contribute to the regulation of HIF-1α, and therefore more knowledge about the PTM landscape of HIF-2α and HIF-3α will be required to increase basic understanding of hypoxia signalling, its crosstalk with other signalling networks, and improve the potential for therapeutic intervention.

## Figures and Tables

**Figure 1 ijms-22-00268-f001:**
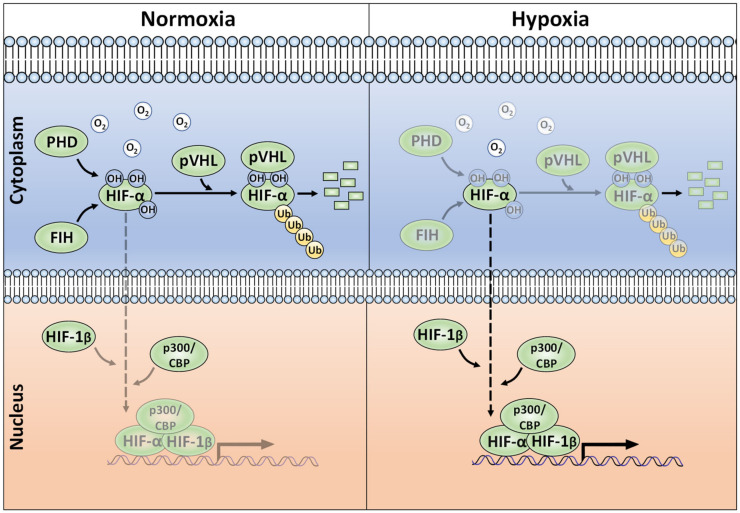
The canonical oxygen-dependent degradation mechanism for HIF-α by pVHL-mediated 26S proteasomal degradation and its inhibition by Factor-Inhibiting HIF (FIH) during hypoxia.

**Figure 2 ijms-22-00268-f002:**
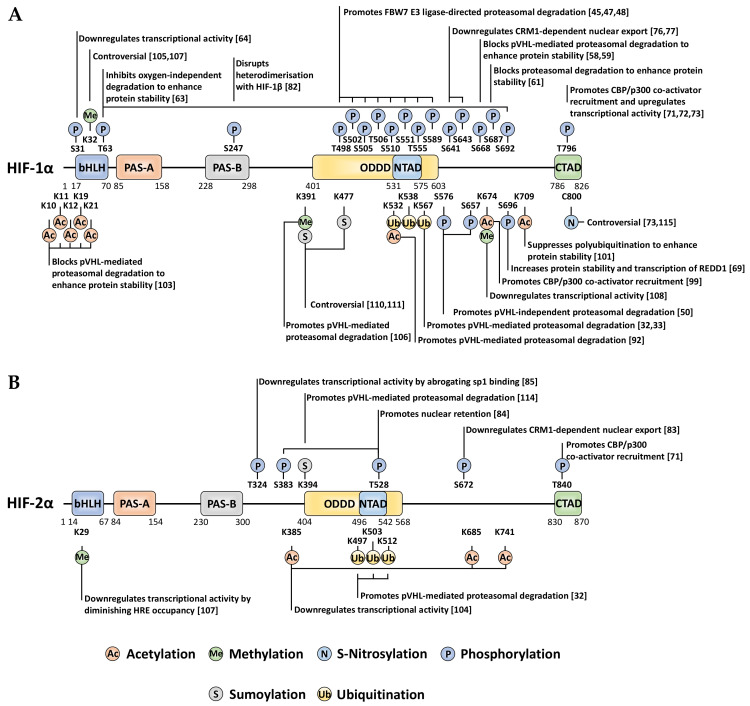
Localisation and function of non-canonical HIF-α post-translational modifications (PTMs) mapped onto full-length HIF-1α (**A**) and HIF-2α (**B**). When a given PTM has multiple publications stating conflicting functional outcomes, then the functionality is denoted as ‘controversial’.

**Figure 3 ijms-22-00268-f003:**
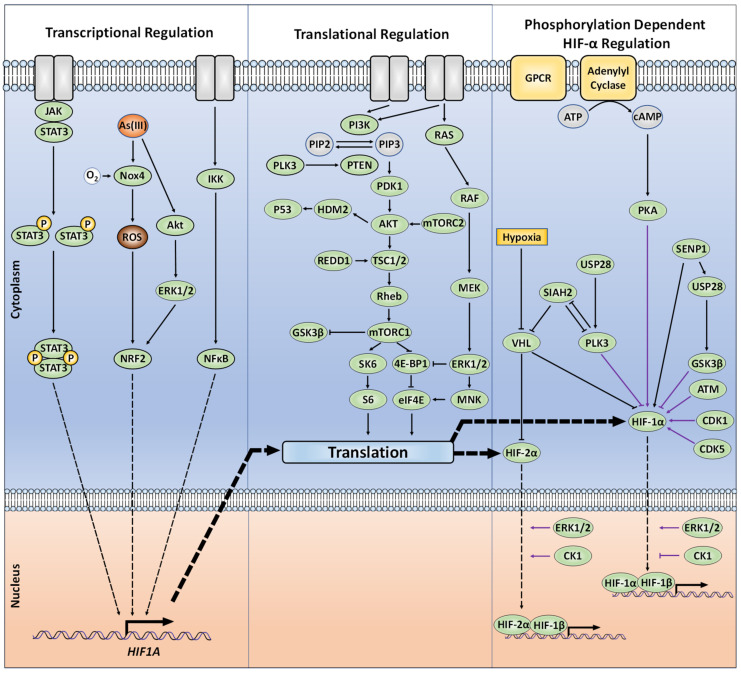
The intracellular protein-dependent cell signalling for the transcriptional, translational, and phosphorylation-dependent regulation of HIF-α. Purple arrows indicate protein kinases that directly phosphorylate HIF-α.

**Figure 4 ijms-22-00268-f004:**
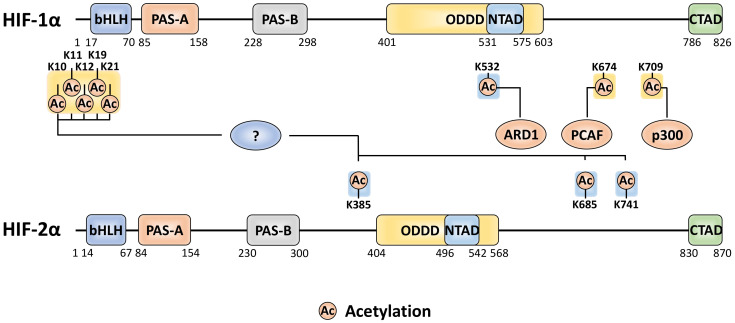
The amino acid sites of HIF-α subjected to acetylation and the acetyltransferase enzymes attributed to the relevant sites. A ‘?’ represents an unknown regulator, not currently described in the literature. PTMs with yellow and blue backgrounds indicate activatory and inhibitory effects, respectively.

**Figure 5 ijms-22-00268-f005:**
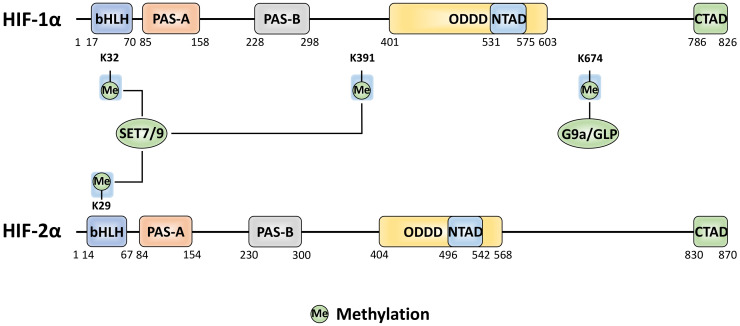
The amino acid sites of HIF-α methylation and the methyltransferase enzymes responsible for each site-specific modification. PTMs with yellow and blue backgrounds indicate activatory and inhibitory effects, respectively.

## Data Availability

Not applicable.

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
