# Peer review of "The Role of Hypoxia-Inducible Factor Post-Translational Modifications in Regulating Its Localisation, Stability, and Activity"

_ijms, 2020, doi:10.3390/ijms22010268_

Round 1

Reviewer 1 Report

This is an excellent review, detailed and well illustrated.

My only comment is about page 2, line 49. Studies have shown that HIF1 and HIF2 can compensate for each other. I have in mind the study my Menrad et al. in Hepatology 2010. The uathors may want to revise the sentence or develop o the subject a little.

Author Response

My only comment is about page 2, line 49. Studies have shown that HIF1 and HIF2 can compensate for each other. I have in mind the study my Menrad et al. in Hepatology 2010. The authors may want to revise the sentence or develop the subject a little.

We have added this information and have balanced the initial sentence: "In addition, some studies showed that neither HIF-1α nor HIF-2α could substitute the lack of DNA binding caused by the absence of the one or the other HIF-α variant [11]. However, others have shown that the loss of a single HIF‐α isoform (either HIF‐1α or HIF‐2α) is compensated by the enhanced expression of the other and promote survival during cancer development [12]." (L49-53 of the revised manuscript)

Reviewer 2 Report

The manuscript by Albanese et al. is a brave attempt to sort through the many posttranslational modifications described for the HIF proteins. Although I'm definitely less an expert in this field as the authors themselves in my opinion they did a quite good job on that. The topic is definitely not an easy one and especially the graphs are carefully arranged and give a nice overview.

My only general comment is that the authors present the current base of knowledge very neutrally. It would be nice if the authors would dare to interpret the presented knowledge more or to emphasize modifications that in their opinion are of crucial biological importance in a pathophysiological context.

An additional chapter on crosstalk between the different PTM would be desirable. 

Minor comments:

Fig. 1: I would suggest to add FIH/CBP/p300 in the graph.

Fig. 2: some lines are fatter than others. 

Fig. 3: Especially in the picture on the right, I would suggest to use different arrows for the actual phosphorylation process, so that it becomes more clear, where it happens and what is just happening downstream.

Some spelling mistakes can be found, e.g. in l. 169 (exerting) and l. 324 (neutralising)

Author Response

My only general comment is that the authors present the current base of knowledge very neutrally. It would be nice if the authors would dare to interpret the presented knowledge more or to emphasize modifications that in their opinion are of crucial biological importance in a pathophysiological context.

An additional chapter on crosstalk between the different PTM would be desirable. 

We agree that the presentation is quite neutral as it is quite difficult to discern between all the modifications, determined in different cells and specific context, the ones standing out. We have added a section in the conclusion, to highlight the pathological implications L428-454 of the revised version.

Minor comments:

Fig. 1: I would suggest to add FIH/CBP/p300 in the graph.

We have done this.

Fig. 2: some lines are fatter than others. 

We realised this as well after submission and have now corrected it.

Fig. 3: Especially in the picture on the right, I would suggest to use different arrows for the actual phosphorylation process, so that it becomes more clear, where it happens and what is just happening downstream.

We have now used purple arrows to indicate direct phosphorylation of HIF-α.

Some spelling mistakes can be found, e.g. in l. 169 (exerting) and l. 324 (neutralising)

Thank you for spotting them. They have been corrected.

Reviewer 3 Report

The manuscript by Albanese et al represents a detailed review on the subject of HIF-α post-translational modifications. This review is detailed, well-written and -organised and covers the wide range and literature of HIF-α modifications in a comprehensive way. As HIF-related biology and regulation is of great interests for a wide scientific audience, this review can be very helpful and informative for the potential readers.

However there are some minor corrections and subjects that must be included prior to acceptance.

1) Figure 1 (Normoxia). Although HIF heterodimer and gene transcription are correctly shadowed, shadowing HIF-β monomer gives the impression that in normoxia HIF-β is not present (It must point somehow that HIF-β is expressed - as it function with other factors e.g. AhR).

2) Figure 2 (A) : S247 phosphorylation should read: "disrupts heterodimerization...."(as correctly stated in text) and not "promote....:"(which is erronneous)

3) Lines 210-215 describing C-terminal HIF-a phosphorylation. It should be also noted that the same T796 phosphorylation abolishes FIH hydroxylation as shown with peptides and structural data (Lancaster 2004 Biochem. J. & Cho et al 2007 FEBS Letters)

4) Section 3.7. Although indirect action of kinases are described here and the section is small I think it should be included that HIF-1A transcription by STAT3 is also controlled by PKR kinase (Papadakis et al 2010 Cancer research) and that CDK8 (although not via its kinase activity) associates with HIF-1α and stimulates RNApol II transcription (Galbraith et al, 2013 Cell).

5) Figure 3. Although CK1 activity is inhibitory for HIF-1 heterodimer (and thus correctly symbolized in figure 3), this is not the case for HIF-2. CK1 facilitates HIF-2α retention to the nucleus and thus acts as activator (symbol in figure must change for CK1 and HIF-2α).

Minor

- In some cases spaces or different fonts are shown (e.g. line 76 HIF-α, unnecessary space; line 229, GSK-3β, β is with different fonts e.t.c)

- Is it possible to illustrate if modifications shown in figures 3 and 4 are activating or not?

Author Response

1) Figure 1 (Normoxia). Although HIF heterodimer and gene transcription are correctly shadowed, shadowing HIF-β monomer gives the impression that in normoxia HIF-β is not present (It must point somehow that HIF-β is expressed - as it function with other factors e.g. AhR).

This is correct, thank you for spotting it. We have modified it and removed the shadow for the HIF-1β monomer.

2) Figure 2 (A) : S247 phosphorylation should read: "disrupts heterodimerization...."(as correctly stated in text) and not "promote....:"(which is erronneous)

Correct. We have corrected this mistake in the figure.

3) Lines 210-215 describing C-terminal HIF-a phosphorylation. It should be also noted that the same T796 phosphorylation abolishes FIH hydroxylation as shown with peptides and structural data (Lancaster 2004 Biochem. J. & Cho et al 2007 FEBS Letters)

We have now added this information: “Interestingly, this T796 phosphorylation may also abrogate the HIF-1α/FIH interaction to enhance HIF-1 transcriptional activity [72,73]”

4) Section 3.7. Although indirect action of kinases are described here and the section is small I think it should be included that HIF-1A transcription by STAT3 is also controlled by PKR kinase (Papadakis et al 2010 Cancer research) and that CDK8 (although not via its kinase activity) associates with HIF-1α and stimulates RNApol II transcription (Galbraith et al, 2013 Cell).

We have added this new information l262-265 and l268-270 of the revised manuscript.

5) Figure 3. Although CK1 activity is inhibitory for HIF-1 heterodimer (and thus correctly symbolized in figure 3), this is not the case for HIF-2. CK1 facilitates HIF-2α retention to the nucleus and thus acts as activator (symbol in figure must change for CK1 and HIF-2α).

This has now been corrected

Minor

  • In some cases spaces or different fonts are shown (e.g. line 76 HIF-α, unnecessary space; line 229, GSK-3β, β is with different fonts e.t.c)

This has been corrected.

  • Is it possible to illustrate if modifications shown in figures 3 and 4 are activating or not?

We have now added yellow and blue background in fig 4 and 5 to show if the modifications are activating or inhibiting.